# The Circadian Regulation of Nutrient Metabolism in Diet-Induced Obesity and Metabolic Disease

**DOI:** 10.3390/nu14153136

**Published:** 2022-07-29

**Authors:** Lauren N. Woodie, Kaan T. Oral, Brianna M. Krusen, Mitchell A. Lazar

**Affiliations:** Institute for Diabetes, Obesity, and Metabolism, Division of Endocrinology, Diabetes, and Metabolism, Department of Medicine, Perelman School of Medicine, University of Pennsylvania, Philadelphia, PA 19104, USA; woodiel@pennmedicine.upenn.edu (L.N.W.); kaanoral@sas.upenn.edu (K.T.O.); brianna.krusen@pennmedicine.upenn.edu (B.M.K.)

**Keywords:** circadian rhythms, obesity, molecular clock, metabolism

## Abstract

Obesity and other metabolic diseases are major public health issues that are particularly prevalent in industrialized societies where circadian rhythmicity is disturbed by shift work, jet lag, and/or social obligations. In mammals, daylight entrains the hypothalamic suprachiasmatic nucleus (SCN) to a ≈24 h cycle by initiating a transcription/translation feedback loop (TTFL) of molecular clock genes. The downstream impacts of the TTFL on clock-controlled genes allow the SCN to set the rhythm for the majority of physiological, metabolic, and behavioral processes. The TTFL, however, is ubiquitous and oscillates in tissues throughout the body. Tissues outside of the SCN are entrained to other signals, such as fed/fasting state, rather than light input. This system requires a considerable amount of biological flexibility as it functions to maintain homeostasis across varying conditions contained within a 24 h day. In the face of either circadian disruption (e.g., jet lag and shift work) or an obesity-induced decrease in metabolic flexibility, this finely tuned mechanism breaks down. Indeed, both human and rodent studies have found that obesity and metabolic disease develop when endogenous circadian pacing is at odds with the external cues. In the following review, we will delve into what is known on the circadian rhythmicity of nutrient metabolism and discuss obesity as a circadian disease.

## 1. Introduction

Obesity and metabolic diseases are among the most prevalent and costly health issues in the modern world. Obesity is strongly correlated with a wide array of serious comorbidities, such as type II diabetes mellitus (T2DM), non-alcoholic fatty liver disease (NAFLD), cardiovascular disease, and, most recently, COVID-19 severity [1,2,3,4,5]. In the United States, more than 42% of the adult population is currently considered obese [6]. Given that these comorbidities are among the leading causes of premature mortality in the world, the prevention and treatment of obesity is becoming an increasingly important public health imperative [6].

Although obesity and metabolic diseases are multifactorial in their causes and presentation, they are particularly prevalent in populations with external pressures that promote disruptions to circadian rhythmicity [7,8,9]. For example, the Nurses’ Health Studies, which followed more than 140,000 women over several decades to evaluate their risk for chronic diseases, revealed that those working rotating night shifts were more likely to develop metabolic diseases such as T2DM than those with more consistent work schedules [10].

Constant access to food, entertainment, shift work, and trans-time zone flights continues to push the limits of the human circadian system. The word “circadian” comes from the Latin words *circa* meaning “about” and *dies* meaning “day”; thus, a circadian rhythm is defined as a biological pattern that oscillates according to an approximately 24 h time schedule and repeats consistently without the input of external stimuli [11,12]. A number of biological processes exhibit circadian rhythmicity and work to maintain metabolic flexibility and organismal homeostasis in the varying environmental states present across a 24 h day [13,14,15,16,17]. The environment (i.e., food availability, light intensity, ambient temperature, etc.) and the needs of an individual within it fluctuate drastically across 24 h, necessitating a flexible metabolic system capable of adjusting quickly and effectively to external changes.

A hallmark of obesity and metabolic disease, however, is a decreased ability to adapt effectively between metabolic states [18,19,20,21]. Furthermore, when endogenous circadian pacing is at odds with the external environment, this finely tuned mechanism breaks down and has been found to be an important risk factor for a series of disease states including cardiovascular illness, substance use disorder, attention-deficit-hyperactivity disorder (ADHD), and certain psychiatric and neurodegenerative conditions such as dementia, depression, and anxiety in addition to obesity and metabolic disease [22,23,24,25,26,27,28,29,30,31].

Here, metabolic flexibility will be explored as a homeostatic component of the circadian system with particular attention paid to circadian dysfunction in the obese state. More precisely, we aim to provide a review of what is known on the circadian rhythmicity of nutrient metabolism alongside a thorough discussion of obesity as a circadian disease.

## 2. Circadian Biology

The rhythm of all mammals living on the Earth’s surface is dictated by the rotation of the Earth around the Sun. The Sun’s photic signals are transmitted from retinal ganglia to the hypothalamic suprachiasmatic nucleus (SCN) [32,33]. The SCN is a bilateral structure located just above the optic chiasm on the anteroventral aspect of the hypothalamus. It is arranged by neurochemical content into two regions: the dorsal shell, composed largely of arginine vasopressin (AVP) containing neurons, and the ventral core, which primarily contains vasoactive intestinal polypeptide (VIP) neurons [32,33]. The VIPergic core neurons receive light-induced signals from the retina, which resonate through the ventral shell to synchronize activity within the SCN and potentiate rhythmic signals throughout the body [32,33,34]. In this sense, the SCN can be considered the “master” circadian regulator. It earned this distinction due to the observations that: (1) SCN ablation in mice results in arrhythmic behavioral rhythms in the absence of light; (2) SCN ablation renders animals unable to entrain behaviorally to a light schedule; and (3) animals with an ablated SCN, upon transplantation of a rhythmic animal’s SCN, adopted the rhythmic behavioral pattern of the donor [35,36].

At the molecular level, the presence of daylight entrains the rhythms of SCN neurons to a ≈24 h cycle by initiating the transcription and translation of clock genes in a transcription/translation feedback loop (TTFL) (Figure 1). The circadian TTFL is comprised of the proteins BMAL1, CLOCK/NPAS2, PER1-3, CRY1/2 and ROR and REV-ERBα/β, all of which show a remarkably high degree of conservation across evolutionary time [12,13,32,37]. BMAL1, CLOCK/NPAS2, and ROR form the positive arm of the loop (Figure 1a), while PERs, CRYs, and REV-ERBs make up the negative arm (Figure 1b). BMAL1 and CLOCK/NPAS2 form a heterodimer in the cytoplasm and, upon translocation to the nucleus, activate gene expression on *Per*, *Cry*, and *Rev-erb*s. PER and CRY form a complex in the cytoplasm and return to the nucleus to inhibit the transcriptional activity of BMAL1::CLOCK/NPAS2 [12,13,32,37] (Figure 1a). REV-ERBs are constitutive repressors of *Bmal1* and *Npas2* that function by actively inhibiting gene transcription at the ROR response element (RORE) [38,39] (Figure 1b). Therefore, accumulation of the PER::CRY complex and REV-ERBs decreases the levels of BMAL1, CLOCK, and NPAS2. A reduction in the abundance and activity of the TTFL’s positive arm decreases transcription of the negative arm components, thereby disinhibiting expression of the positive arm to restart the TTFL. Furthermore, PER out of complex with CRY will inhibit REV-ERBs’ repression of *Bmal1* [40] (Figure 1b). This series of interlocking regulatory loops creates the stable and precise oscillatory pattern of core clock gene expression.

The downstream impacts of core clock components on clock-controlled genes allow the SCN to coordinate the rhythm for many physiologic, metabolic, and behavioral processes [12,13,34,41]. Indeed, almost half of the mammalian genome exhibits rhythmicity in its pattern of expression [42]. As a result, circadian clocks are thought to function in a hierarchical system in which the SCN is the top-level regulator of light-entrained rhythms. The core clock genes, however, are ubiquitous and oscillate within the TTFL throughout the body while regulating clock-controlled genes that are highly specific and tailored to the function of the tissue in question [37]. Peripheral tissues, in particular the liver and adipose tissue, are highly sensitive to nutritional cues as circadian timekeeping signals [43,44,45,46,47,48]. Therefore, it is difficult to establish a defined hierarchy of circadian control when considering the circadian system and its function in processes other than behavioral rhythmicity. Next, we review those peripheral rhythms in greater detail, specifically in the context of how rhythms propagated by feeding and fasting help maintain metabolic flexibility in tissues outside of the SCN.

## 3. Rhythms of Feeding and Fasting

The rhythms of carbohydrate, lipid, and protein metabolism have been detailed in previous publications in both human and rodent populations [49,50,51,52,53,54,55,56,57]. During the active phase (light cycle for humans (Figure 2); dark cycle for rodents), food consumption results in an increase in blood glucose that elicits increased glucose absorption and glycogen synthesis in the skeletal muscle and liver [58,59,60,61] (Figure 2). Lipids are also absorbed more readily, and lipoprotein lipase activity increases [59,62,63,64] (Figure 2). Amino acid absorption and protein synthesis in the skeletal muscle and liver have also been found to increase during active-phase food consumption [65,66,67] (Figure 2).

Further breakdown of macronutrient metabolism during the active phase reveals that carbohydrates are preferred and more easily metabolized during the early active phase, whereas lipids and proteins are metabolized preferentially during the late active phase [49,51] (Figure 2). The inactive phase (dark cycle for humans (Figure 2); light cycle for rodents), by contrast, is characterized by an increase in catabolic processes [49,68,69]. Glycogenolysis is upregulated in the skeletal muscle and liver, while lipolysis is increased in the adipose tissue [58,59,60,61]. During inactivity, glutamine synthase and autophagy pathways are upregulated in the skeletal and cardiac muscle tissues as well as in the liver [68,70,71] (Figure 2).

The rhythms of metabolism at the macronutrient level closely follow the fed/fasting cycle [72]. Upon a 24 h fast, approximately 80% of oscillating transcripts lose their rhythmicity in the mouse liver [60]. Similarly, core clock gene expression in the livers of mice on a reverse-phase feeding schedule (feeding in the dark and fasting when light is available) phase-shifted by approximately 12 h, indicating a highly interdependent relationship between feeding schedules and the rhythms of peripheral clocks [45,73]. These results imply that peripheral tissues—and especially the liver—may depose the SCN in the rhythm hierarchy when the fed/fasting cycle is disrupted. In fact, an animal’s fed/fasting cycle can partially restore circadian oscillations in peripheral tissue gene expression even in instances of SCN ablation [74,75].

Several components of the core clock TTFL are found at the intersection of rhythmicity and metabolism (Figure 3). A whole-body knockout of *Bmal1* (*Bmal1*-KO) completely eliminates both central and peripheral rhythmicity, resulting in a myriad of health issues outside of behavioral arrhythmicity (e.g., low body weight, reduced lifespan, low activity) [76,77,78]. Recent studies using the *Bmal1*-KO model with a liver-only *Bmal1* reconstitution (Liver-RE) found that restoring hepatic *Bmal1* expression led to rhythmic oscillations in both the hepatic core clock and carbohydrate metabolism genes. Body weight, lifespan, and hepatic liver metabolism, however, were not rescued by Liver-RE, indicating the importance of *Bmal1* in the full circadian system [79,80].

Similarly, CRYs have been found to interact with nuclear receptors outside of the circadian system to modulate metabolism and impose circadian rhythmicity on the activity of several nuclear receptors. In a study specifically examining glucocorticoid receptor (GR), CRYs were found to inhibit GR expression. *Cry* deficiency resulted in glucose intolerance and constitutively high levels of corticosterone [81]. CRY1 and 2 also interact with PPARd to modulate lipid metabolism and exercise capacity [82].

The REV-ERBα and β nuclear receptors are also key regulators of rhythmic metabolism. In the molecular circadian clock, REV-ERBα and β compensate for each other, but REV-ERBα appears to be the more dominant isoform [83]. It has been shown to repress several genes involved in metabolic homeostasis including the gluconeogenic gene glucose 6-phosphatase [84] and apolipoprotein C-III (apoC-III), which regulates triglycerides and very low-density lipoproteins [85,86]. The total loss of REV-ERBs disrupts rhythmic gene expression to an extent similar in scope to other core clock double knockouts such as *Cry1*/*2* and *Per1*/*2* [87].

More recent studies have evaluated tissue-specific roles of REV-ERBs in metabolism. The hepatocyte-specific knockout of *Rev-erbα* and *Rev-erbβ* resulted in disruptions to liver lipid metabolism, leading to extreme hepatic steatosis and significant dampening of metabolic rhythms [45]. By contrast, an SCN-specific *Rev-erbα*/*β* double knockout shifted metabolic and behavioral rhythms by approximately three hours, entraining affected mouse models to a roughly 21 h day. SCN *Rev-erbα*/*β* deletion also rendered mice significantly more susceptible to acute weight gain when fed a high-fat diet [24], indicating that REV-ERBs play an important role in the prevention of diet-induced obesity. Importantly, readjusting the light/dark cycles of affected mice to their endogeneous 21 h clock appeared to rescue the adverse metabolic effects of the SCN *Rev-erβ* double knockout [24]. This result highlights the costly metabolic consequences that occur when there is misalignment between organismal circadian rhythms and environmental light/dark cues. Due to the impact of REV-ERBs on both *Bmal1* and metabolic homeostasis, they are considered key regulators at the intersection of circadian rhythmicity and metabolism—especially as it relates to feeding and fasting.

## 4. Time-Restricted Feeding/Eating

Time-restricted feeding (TRF) in rodents or time-restricted eating (TRE) in humans has been shown to have positive impacts on metabolic health in both animal and human models (Table 1) [43,44,46,88,89,90,91]. TRF works to synchronize the timing of food intake with the natural circadian rhythms of nutrient metabolism. It induces a rhythmic schedule of metabolic flexibility by enforcing distinct feeding and fasting schedules over 24 h. Studies utilizing TRF find that restricting food consumption to ≈9 h windows during an organism’s active phase can improve glucose metabolism, insulin signaling, and reduce markers of NAFLD [43,44,46,88,92].

While the phenotypic effects of TRF are encouraging, it is worth noting that clock genes within the hypothalamus and SCN such as *Rev-erbα*, *Bmal1*, and *Per2* are resistant to diet-induced changes and have been shown to remain rhythmic in both normal chow- and Western diet (WD)-fed animals [24,47,95]. Furthermore, applying reverse phase feeding (RPF) to the inactive phase shifts *Per1*/*2* expression in peripheral tissues as well as in a variety of other brain regions, but not in the SCN [96]. These studies indicate that the SCN is resistant to nutritional changes and imply the presence of one or more food-entrainable oscillators (FEOs) outside of the SCN. As of this writing, the liver is the most viable location for an FEO because, even under dark:dark light conditions, 15% of the hepatic transcriptome retains circadian oscillation [60].

Ever since the presence of one or more FEOs was suggested, many research initiatives have been launched to pinpoint their location and better understand the mechanisms by which they function. TRF studies have been the most popular way to examine cell autonomous FEOs. Moreover, TRF has repeatedly been examined as a chronotherapeutic alternative to traditional restrictive dieting to improve metabolic health. Hatori et al. published one of the seminal large-scale physiologic and metabolic TRF studies. Mice were given either standard rodent chow or a high-fat diet (60% fat/kcal) and were allowed only eight hours of food access during the active period. The restricted animals consumed an equivalent number of calories as their ad libitum counterparts, but time-restricted HFD animals were protected against the metabolic effects of the diet. TRF preserved the rhythmic transcription of *Per2*, *Bmal1*, *Rev-erbα*, and *Cry1* in the liver of high-fat fed mice while protecting against diet-induced damage to glucose and lipid homeostasis [43].

A subsequent study from the same group examining several different feeding paradigms and TRF schedules ranging from 8 to 12 h of restriction found that a TRF schedule less than 12 h was the most effective at protecting against weight gain and metabolic disruption [44]. An 8–9 h TRF schedule was effective at protecting against weight gain, inflammation, hyperglycemia, and hyperinsulinemia in both high-fat and high-fructose fed mice. TRF also improved the metabolic profile of high-fat and high-fructose fed animals, with clear rhythmic separation of transcripts and metabolites implicated in fed and fasted states. However, TRF does not appear likely to have a substantial “legacy effect”—the idea that a treatment may continue to be beneficial even after it is stopped. Upon cessation of a strict TRF regime, associated metabolic benefits were only observed to last two days as animals quickly adopted the ad lib gene signature, losing the protection previously conferred by TRF [44].

Other labs have corroborated many of these findings but with varying degrees of effect on body weight. A 2010 study compared a 12 h TRF to an 8 h TRF and found that the 8 h TRF was most effective in improving metabolic health. In contrast to the above-mentioned findings from Hatori et al. and Chaix et al., the 8 h TRF program was not found to decrease body weight or body fat [88]. A 2018 study examining TRF’s reversal potential instead of its protective effects found that body weight and body fat did not significantly decrease after either 4 or 10 weeks on an 8 h TRF schedule [46]. Importantly, however, TRF improved markers of metabolic flexibility as well as hepatic triglyceride content, insulin resistance, and glucose intolerance, demonstrating that aligning the rhythms of nutrient metabolism with nutrient availability can improve metabolic health independent of weight loss [46].

A 2017 study found that meal timing was also able to shift metabolic rhythms in humans. A 5.5 h shift in mealtime resulted in a significant phase delay in circulating glucose in healthy young men. Meal shifting did not alter circulating insulin or triglycerides, but it did cause a phase delay of *Per2* in white adipose tissue. All participants retained normal sleep patterns, indicating that, as in rodents, the human SCN is resistant to meal timing [93]. In 2018, a TRE paradigm was applied to pre-diabetic men to explore its utility as a clinical approach for improving metabolic health. Here, they found that individuals consuming all their meals before 3 p.m. (early TRE, eTRE, ≈6 h) had significantly improved insulin sensitivity, pancreatic beta cell responsivity, blood pressure, and markers of oxidative stress. The eTRE group did not lose weight when compared to individuals allowed to eat for a full 12 h, meaning that, as in mice, the metabolic benefits of TRE are independent of weight loss [91]. In a subsequent study from the same lab on another cohort of pre-diabetic men and women, they found that eTRE improved the intensity of glycemic excursions and suggested that eTRE may increase autophagy to promote anti-aging effects [94].

There exists more evidence behind the claim that time-restricted feeding is best suited for individuals already suffering from metabolic dysfunction. A 2020 study by Wilkinson et al. showed that TRE led to sharp decreases in fasting blood glucose levels, especially among individuals who were already diabetic or pre-diabetic. TRE also rendered individuals less susceptible to large fluctuations in blood glucose following a high-carbohydrate meal. The same study found that restricting caloric intake to a 10 h feeding window over 12 weeks led to significant decreases in body weight and waist circumference among patients who had previously been diagnosed with metabolic syndrome—greater than the TRE-induced weight loss experienced by healthy individuals. Although similar weight loss results were obtained via standard dieting strategies, simple caloric restriction led to significantly smaller reductions in blood pressure and serum lipid levels compared to TRE [89]. However, recent findings suggest that caloric restriction on a time-restricted feeding schedule can improve well-being and increase lifespan to a greater extent than either caloric restriction or time-restricted feeding alone [97].

As promising a therapeutic intervention as time-restricted feeding/eating may appear to be, larger studies (especially in humans) and more robust regulation of TRF/TRE protocols is necessary before it can become a reliable, mainstream treatment for obesity. Closer investigation into the effects of TRF/TRE on specific patient populations and on other disease mechanisms (neurological, psychiatric, and cardiovascular) is needed to better interrogate and eventually understand its full therapeutic potential. In any case, the apparent viability of TRF/TRE as a treatment for obesity remains an exciting avenue for future research in the field.

## 5. Obesity as a Circadian Disease

The modern world is active for all 24 h of the day. Constant access to food, entertainment, shift work, and trans-time zone flights push the limits of human circadian physiology. The circadian system has evolved to temporally separate cellular states and promote flexibility in metabolic responses to the different environmental demands of a 24 h day (Gerhart-Hines and Lazar, 2015). However, when endogenous circadian pacing is at odds with the light:dark cycle, a myriad of health issues—collectively known as circadian time-sickness—can occur [26].

Disrupted sleep is a major contributor to circadian time-sickness and metabolic dysfunction. Indeed, just one night of sleep disturbance impairs glucose metabolism in humans [27,28,29]. Individuals who go to sleep later and sleep for fewer hours are more likely to be obese than people who regularly obtain a quality night’s sleep [98,99]. Decreased sleep quality and quantity also increases the risk of developing T2DM [30,31,98]. Sleep disturbances elevate orexin-mediated sympathetic nervous system activity, which causes elevated gluconeogenesis and can lead to glucose intolerance and insulin resistance [100]. Importantly, melatonin expression is constitutive and does not oscillate in a circadian manner among individuals with diabetes or obesity, further suggesting that the rhythms of sleep and metabolism are closely interconnected with one another [101,102,103,104]. It follows, then, that exogenous melatonin administration in humans can prevent weight gain, hyperglycemia, hyperinsulinemia, and hyperlipidemia [101,102,103,104].

Diet can impact circadian health even without significant alterations in sleep quantity or quality. As discussed earlier, all animals including humans alternate between periods of feeding and fasting. One of the major evolutionary theories behind the architecture of the circadian clock is that it serves to establish temporal separation between those metabolic states. This theory is supported by the breakdown of rhythmicity and metabolic flexibility, as well as the decline in overall health, that results when distinct fed/fasting states are not maintained [105].

In chow-fed mice, reverse-phase feeding (RPF) to the inactive phase results in elevated triglycerides, altered glucose metabolism, and weight gain when compared to TRF mice fed in their active phase [106]. This phenotype was exacerbated when an inactive phase shift work paradigm was coupled with RPF. Notably, however, TRF was able to correct the metabolic consequences of shift work during the inactive phase. Inactive workers on active TRF had corrected rhythms in activity, blood glucose, and triglyceride levels with significantly reduced body weight [106].

A 2016 study by Yasumoto et al. found that RPF also increased plasma corticosterone, insulin, and leptin levels [107]. At the molecular level, RPF produced phase delays or shifts in core clock genes in the liver, white adipose tissue, and skeletal muscle [107]. This is consistent with more recent work by Guan et al., which showed that the implementation of an RPF regime led to approximately 12 h phase shifts in the expression of core clock genes such as *Reverbα*/*β*, *Bmal1*, and nearly all oscillating genes in the liver [45]. Altogether, the close alignment between feeding patterns and circadian gene expression indicates that feeding is a crucial timekeeper for peripheral clocks such as the liver. More broadly, the metabolic consequences of RPF demonstrate that uncoupling food consumption from the body’s natural internal clock is detrimental to overall health.

A 2019 study from Panda’s group found that the liver-specific deletion of critical core clock genes such as *Rev-erbα/β* and *Bmal1*, together with whole-body knockouts of *Cry1/Cry2*, led to extreme and rapid weight gain in mice, once again showing that an intact circadian system is critical to the maintenance of metabolic homeostasis [92]. However, the implementation of TRF lessened the phenotypic consequences of missing core clock machinery. TRF mice were significantly more resistant to accruing excess weight even in the absence of these key core clock components. Broadly speaking, their findings demonstrate that the rhythms of feeding and fasting may be capable of reversing the gene expression changes caused by absent core clock elements [92].

Natural circadian oscillations can also mitigate the metabolic consequences associated with a high-fat diet. An SCN-specific deletion of both *Rev-erbα* and *Rev-erbβ* shifted their internal circadian clocks by around three hours; they found that double knockout mice were also significantly more vulnerable to HFD-induced weight gain than mice whose REV-ERB nuclear receptors remained intact [24]. Other studies observed a similar phenotypic effect upon deletions or mutations of core clock genes [108]; for example, *Clock* mutations altering internal mouse clocks also rendered mice more susceptible to weight gain and hyperleptinemia [109,110]. In both cases, a desynchrony between organismal endogenous clocks and environmental light cues exacerbated metabolic dysfunction. Appropriately functioning core clock machinery is therefore integral to the maintenance of metabolic homeostasis. Framed slightly differently, misalignment between circadian gene expression and external light/dark cycles hinders bodily protection against the metabolic effects of high-fat diets (such as diet-induced obesity).

While disruptions to the rhythmic expression of core clock genes leads to phenotypic changes upon exposure to a high-fat diet, the causal relationship appears to swing in both directions—that is, high-fat food consumption has also been found to alter and dampen patterns of circadian gene expression [44,47,95]. Indeed, the rhythmicity of *Clock*, *Bmal1*, and *Per2* gene expression was significantly dampened in the liver and white adipose tissue among mice on a high-fat diet [44,47,95]. HFD has important ramifications regarding metabolic flexibility, making it more difficult for an organism to switch between carbohydrate and lipid metabolism [44,46,47]. These results combine to show that patterns of rhythmic gene expression are key mediators for a wide variety of metabolic processes. Molecular disruptions to core clock machinery appear to increase the propensity for a wide array of metabolic disease states such as obesity and T2DM. Beyond genetics, dietary choices such as a high-fat diet threaten to upend the body’s natural circadian rhythms, creating a self-reinforcing cycle of metabolic dysfunction that can have serious health consequences.

## 6. Conclusions

Examination of the literature reveals that the circadian clock plays a crucial role in the regulation of metabolic performance. In this review, we have used a metabolism-focused lens to explore the intersection of circadian biology and the pathogenesis of prevalent diseases such as obesity and type II diabetes. As the interplay between core clock machinery and metabolism becomes clearer, this relationship appears poised to offer a variety of novel therapies to combat metabolic dysfunction.

Indeed, the circadian clock may be harnessed to treat human disease. In addition to time-restricted feeding, several other chronotherapeutic interventions may have promising clinical applications. The time-specific administration of certain zero-calorie sweeteners (which activate proteins promoting glucose metabolism and appetite suppression) before high-carbohydrate meals has been found to have corrective properties against hyperglycemia and diet-induced obesity. It is important for the possible metabolic benefits of chronotherapy to extend beyond feeding patterns. For instance, the chronic provision of orexin agonists was much more effective at improving glucose tolerance when administered during sleep than when animals were in their active phase [111]. Although these results support the hypothesis that the timing of drug administration matters, chronotherapy remains a new area of inquiry for the field, and much more work will be required before chronotherapy can become a widely accessible, standard-of-care aspect of treatment of human illness. However, the rapid pace at which researchers are uncovering the role of the circadian clock in complex disease mechanisms suggests that the future of chronotherapy is brighter than ever before.

## Figures and Tables

**Figure 1 nutrients-14-03136-f001:**
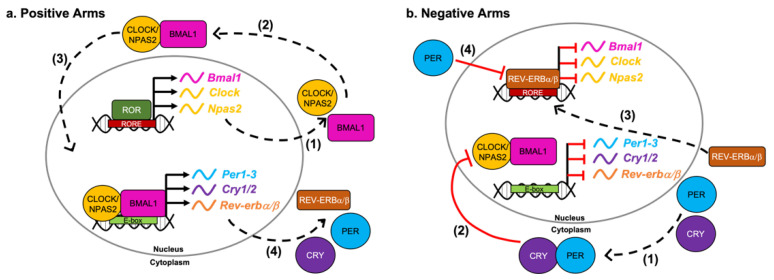
The Circadian Transcription–Translation Feedback Loop (TTFL). (**a**) Positive arms of the TTFL (1) ROR binding to the RORE initiates transcription of *Bmal1*, *Clock* and *Npas2*, which are shuttled to the cytoplasm for translation. (2) BMAL1 and CLOCK/NPAS2 form a complex in the cytoplasm. (3) The BMAL1:CLOCK/NPAS2 complex translocates to the nucleus where it binds to E-box elements to (4) upregulate transcription of *Per*, *Cry* and *Rev-erbα*/*β* and other core clock genes; (**b**) Negative arms of the TTFL (1) PER and CRY form a complex that (2) translocates to the nucleus and inhibits the transcriptional activity of BMAL1::CLOCK/NPAS2. REV-ERBα/β translocates to the nucleus and represses the transcription of *Bmal1*, *Clock* and *Npas2*. (4) PER monomers inhibit REV-ERBα/β.

**Figure 2 nutrients-14-03136-f002:**
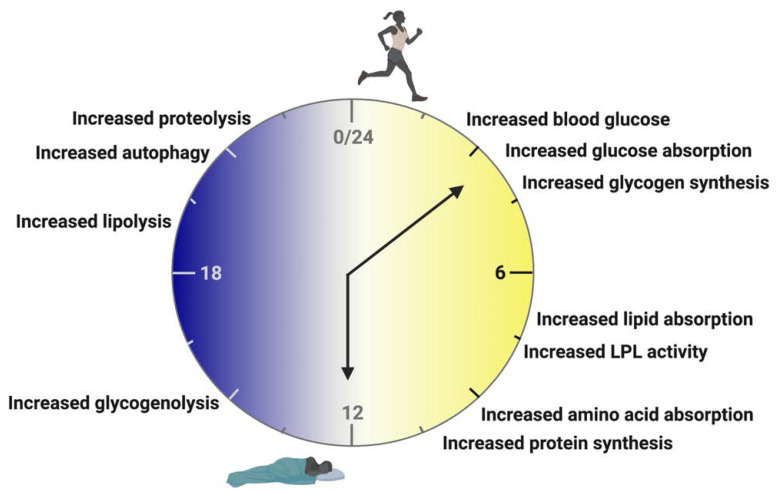
The Human Metabolic Clock. During the active phase, food consumption results in an increase in blood glucose that elicits an increase in glucose absorption and glycogen synthesis. Lipids are also absorbed more readily, and lipoprotein lipase (LPL) activity increases. Amino acid absorption and protein synthesis increase during active-phase food consumption. Carbohydrates are more easily metabolized during the early active phase, whereas lipids and proteins are metabolized preferentially during the late active phase. The inactive phase is characterized by an increase in catabolic processes. Glycogenolysis is upregulated and lipolysis is increased. During inactivity, glutamine synthase and autophagy pathways are upregulated. Created with BioRender.com (accessed on 28 June 2022).

**Figure 3 nutrients-14-03136-f003:**
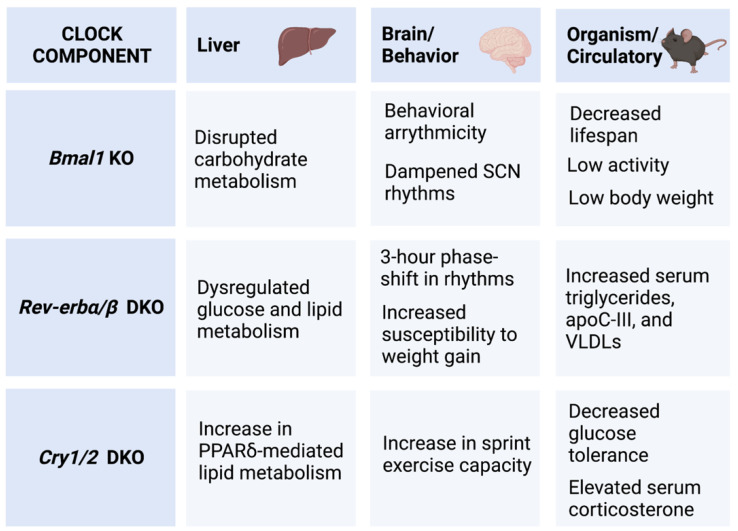
Select Effects of Molecular Clock Knockouts. Examples of select metabolic effects arising from molecular clock knockouts. Created with BioRender.com (accessed on 28 June 2022).

**Table 1 nutrients-14-03136-t001:** Selected Time-Restricted Feeding/Eating Studies and their Effects on Metabolic Health.

Study	Organism	Timing of TRF/TRE	Length of TRF/TRE	Effects on Metabolism and Health
Bray et al. International Journal of Obesity 2010 [88]	FVB/N mouse	8 h	12 weeks	No protection against HFD-induced weight gain, but improved glucose and lipid metabolism
Hatori et al. Cell Metabolism 2012	C57Bl/6J mouse	8 h	17 weeks	Protected against HFD-induced disruptions in glucose and lipid metabolism
Chaix et al. Cell Metabolism 2014 [44]	C57Bl/6J mouse	8–12 h range	12–36 weeks	8–9 h range protected against diet-induced weight gain, inflammation, hyperglycemia, hyperinsulinemia, and disruption in metabolite flux
Wehrens et al. Current Biology 2017 [93]	Healthy human males	5 h in late active phase	6 days	Shift in adipose Per2 expression, preserved behavioral activity, no added metabolic benefit of TRE for healthy human males
Woodie et al. Metabolism 2018 [46]	C57Bl/6N mouse	8 h	4 and 10 weeks	No protection against HFD-induced weight gain, but metabolic flexibility, insulin and glucose tolerance, and hepatic steatosis
Sutton et al. Cell Metabolism 2018 [91]	Pre-diabetic human males	6–7 h in early active phase	5 days	Improved insulin sensitivity, pancreatic beta cell responsivity, blood pressure, and markers of oxidative stress
Jamshed et al. Nutrients 2019 [94]	Pre-diabetic human males and females	7 h in early active phase	4 days	Improved glycemic excursions and increased markers of autophagy and anti-aging
Wilkinson et al. Cell Metabolism 2020 [89]	Human males and females with metabolic syndrome	10 h	12 weeks	Decreased body weight, blood pressure, cholesterol and A1C while improving sleep quality

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
