# Peer review of "The Circadian Regulation of Nutrient Metabolism in Diet-Induced Obesity and Metabolic Disease"

_nutrients, 2022, doi:10.3390/nu14153136_

Round 1
Reviewer 1 Report
Dear authors,
The manuscript entitled "The Circadian Regulation of Nutrient Metabolism in Diet- 2 Induced Obesity and Metabolic Disease" addresses a very important and little-explored theme in the scientific community in relation to the discussion of shift work and jet-lag effects. The review was very well directed, presenting figures and tables with high relevance to the article. I just ask you to remove the description from table 1 that was duplicated on page 10 lines 439-440.
Reviewer 2 Report
The article: The Circadian Regulation of Nutrient Metabolism in Diet-Induced Obesity and Metabolic Disease, it is very interesting because discuss obesity as a circadian disease.
